# The Effect of Nordic Walking Training with Poles with an Integrated Resistance Shock Absorber on the Body Balance of Women over the Age of 60

**DOI:** 10.3390/healthcare9030267

**Published:** 2021-03-02

**Authors:** Katarzyna Marciniak, Janusz Maciaszek, Magdalena Cyma-Wejchenig, Robert Szeklicki, Rafał Stemplewski

**Affiliations:** Department of Physical Activity Sciences and Health Promotion, Poznań University of Physical Education, 61-871 Poznań, Poland; jmaciaszek@awf.poznan.pl (J.M.); magdalenacyma@gmail.com (M.C.-W.); szeklicki@awf.poznan.pl (R.S.); stemplewski@awf.poznan.pl (R.S.)

**Keywords:** physical activity, Nordic walking, Romberg sharpened test, posturographic evaluation, functional training, aging, body balance

## Abstract

The aim of this study was to assess the changes in the balance of healthy, elderly women as a result of Nordic walking training using of two types of poles: classic poles, and poles with an integrated resistance shock absorber. Thirty-four women completed the experiment (Age = 66.8 ± 4.1 years). They were randomly assigned into the experimental group—training with poles with an integrated resistance shock absorber—EG (*n* = 17), and the control active group—training with classic poles—CG (*n* = 17). Body balance was measured with Romberg sharpened test and using the AccuGait ™ force platform before and after an intervention lasting for 8 weeks (2 training sessions × 75 min per week). In ANOVA analysis, the Romberg sharpened test showed a statistically significant effect of “time” (F = 18.62, *p* < 0.001, ɳ^2^ = 0.37). No interaction effect or clear “time” effect was observed in the ANOVA analysis for the results of posturographic tests (*p* > 0.05). The lack of pre-post differences in posturographic tests indicates that further research is necessary, where, e.g., there are extended intervention times and more difficult examination tasks are performed. In the case of the Romberg test, both groups improved their results, which may indicate an improvement in perception and processing of sensory information, regardless of the type of training applied.

## 1. Introduction

The level of body balance depends on the feedback from the following three systems: visual, proprioceptive, and vestibular [1,2].

Due to physiological changes linked to the aging process of the body, the deterioration of all elements influencing the maintenance of proper body balance can be noticed [3].

Deficiencies in maintaining a stable upright posture during daily activities are a major risk factor contributing to falls in elderly people [4]. Falls, along with their consequences, pose a serious threat to seniors’ health or even life. According to Sample et al. [5] they occur in up to 33% of the elderly. A serious injury, which is a common aftereffect of a fall, can result in permanent loss of the senior’s independence from others. With age, the ability to perform complex motor tasks decreases. This change also applies to the gait. Even in elderly people with good health and bodily functions a decrease in walking speed and step length, as well as a shortened transfer phase can occur; thereby, the length of the double support phase increases [6]. When there is a risk of falling, the elderly limit their cognitive functions by focusing on keeping their balance [7]. Moderate physical activity (PA) is highly recommended for seniors, as it is crucial for maintaining good health [8,9,10,11,12,13]. According to Yorston et al. [14], there is a significant positive link between taking up physical activity and maintaining fitness and independence in adults. Therefore, various forms of PA willingly undertaken by the elderly are still being sought, as they are known to improve their overall functional efficiency, including body balance.

Previous studies demonstrated positive effects of various forms of physical training on the gait and body balance [15,16,17]. One of the frequently examined forms of physical activity is Nordic Walking (NW). Research confirms that NW can be more effective than standard walking in preventing changes in gait quality, including reduced walking speed [18,19]. Some benefits of NW include: an improvement in physical fitness [20], muscle strength [21,22,23], and body balance [20,23,24]. Just after 8–12-week cycles of NW interventions, there was an improvement in static balance in seniors, both in functional tests [23] and in tests on a posturography platform [24,25,26]. However, so far, the NW influence has been examined with the use of standard poles.

A relatively new form of NW is marching with poles containing a resistance shock absorber (RSA). Such training combines aerobic exercise with strength training of the upper limbs and upper torso muscles [27]. The American College of Sports Medicine (ACSM) emphasizes the necessity of strength training in seniors as one of the pivotal elements for comprehensive management to maintain the appropriate level of physical fitness in the elderly [28]. People walking with RSA poles do not have stable support. This situation is similar to exercises on unstable surfaces which are commonly used to improve the balance of the body in rehabilitation [29]. RSA poles bend when walking, this movement may seem to be moving on unstable ground. We assumed the dynamics of walking cause sudden changes in the position of the whole body, including the limbs, necessitating the reflex stabilization of the joints, which may result in improved proprioception and increased level of balance in exercisers.

The aim of this study was to assess the changes in the balance of healthy, elderly women as a result of NW training using of two types of poles: classic and RSA. We hypothesized that training with RSA poles has a much more positive impact on body balance than training with classic poles.

## 2. Methods

### 2.1. Study Design

A randomized parallel-group study design was used with a ratio of 1:1. Two groups (experimental and control) took part in 8-weeks (twice a week, 75 min) PA intervention. Nordic walking with two different kinds of poles was used. Before and after intervention body balance was assessed. The assessors who did measurements were blinded (M.J.; C-W.M.; R.S.^1^). The main researcher (K.M.) who performed the intervention and the researcher calculating the results (R.S.^2^) were excluded from the measurements.

A detailed study scheme is presented in Figure 1.

### 2.2. Participants

Initially, 50 women aged 60–75 were recruited for the study. Potential participants were recruited from local senior centers, through ads in the local press and via social media. The following inclusion criteria were taken into account: age over 60; good verbal and intellectual contact; lack of contraindications to moderate exercise. Subjects were qualified to participate in the project based on medical history and cardiology examination. Each person was free from diseases that could directly affect the gait stereotype (e.g. vestibular neuritis, multiple sclerosis). Persons who had one of the factors were excluded: dizziness, diabetes 2, obesity, and drugs that lower blood pressure, disease of the musculoskeletal system of the lower limbs and spine. Finally, 40 women over 60 years of age participated in the experiment (Age = 66.8 years, age range: 60–75years).

The study was approved by the local Bioethics Committee in Poznań No. 1041/18. in line with the Declaration of Helsinki, and the participants were asked to give their informed written consent to take part in the research. All precautionary measures were taken to protect the participants’ privacy and the confidentiality of their personal data. Furthermore, each participant was notified of their right to refuse to participate in the study or to withdraw their consent at any time, without any consequences. The subjects familiarized themselves with the experimental procedure along with all aspects of the study, such as: objectives of the experiment, methods, test sequence, risks associated with the experiment and expected benefits, as well as inconveniences it may cause.

### 2.3. Intervention

The experiment lasted 8 weeks, with sessions twice a week (16 sessions), which was sufficient to notice improvement in motor abilities [30]. All the women participated in the training sessions at the same time. The women from the EG group used poles with an integrated shock absorber with a resistance force of 4 kg (Slimline BungyPump, Sport Progress International AB, Västernorrland, Sweden), while women from the CG group walked with classic NW poles. Before the intervention, the participants familiarized themselves with the equipment and the correct marching technique during the 60-min training session. The training took place in a city park, in the morning time.

After 10–15min of warm-up, both groups (EG, CG) walked along the inside lanes of the park, on the varied ground. During the whole intervention, the distance was gradually increased (from 3.5 to 4.5 km) with a speed of about 6 km/h. The length and time of the route was measured using the Endomondo app [31]. After half of the planned distance of walking participants did resistance exercises and balance training (15 min). The training plan was compliant with the American College of Sports Medicine (ACSM) recommendations for adults and healthy people: one set of 8–10 exercises for major muscle groups at least 2 days a week with 8–12 repetitions for each exercise [32].

During each session, the participants performed a set of exercises, which consisted of: exercises of the main muscles (warm-up), during which vigorous swings of arms, legs, torso twists, bends, squats, etc., were performed (pace was adapted to the participants’ abilities). Then, in the main part, after half of the distance, everyone performed equivalent exercises, where the position for exercises was standing on one leg with different variants of body position and movement, e.g., tilting the torso, bending and moving the raised leg forward, sideways, etc. In the end, static stretching of the main muscles was performed, i.e., the quadriceps muscle of the thigh, gluteal muscles, calves with stretching of the Achilles tendons, as well as the muscles of the neck and upper limbs. 

The minimum attendance was assumed at the level of 80%, which required participation in at least 13 training sessions [27]. The trainer (K.M.) had the required qualifications as an instructor and trainer (International Nordic Walking Association).

### 2.4. Walking Technique with the Use of Poles

The technique of walking with poles (classic and RSA) consists of a cyclical, repetitive and alternating movement of the whole body. During the walk, not only the muscles of the limbs are activated, but also the muscles of the torso. Locomotion with the use of poles begins with coordinated movement of one hand being stretched forward, with the heel of the opposite foot placed on the ground at the same time. The hand of the upper limb stretched forward is clamped on the handle of the pole gripping it with all the fingers (from I to V), which automatically positions the pole obliquely to the ground (about 45°). In this phase, not only the upper limb is engaged, but also the entire shoulder girdle, because the forward move of the hand takes place while the scapula is extended forward causing the torso to be placed in a rotating position.

The next move is bouncing off the ground using the pole with the simultaneous rolling of the foot from the heel to the toes (propulsion) and pushing the lower limb from the forefoot, followed by the transfer of the prop to the opposite side with the accompanying rotation of the torso to the opposite side. As a result of acceleration, there is a noticeable increase in the work of the limbs and torso, and the body is tilted forward with the center of gravity being shifted forward.

The whole process is alternating concentric and eccentric work of the opposing muscle groups of the upper limbs, lower limbs and torso.

RSA poles have a built-in shock absorber with a total length of 20 cm; therefore, the positioning of the upper limbs while marching with the RSA poles is different than with NW poles, as shown in Figure 2.

When walking with RSA poles, exercisers do not have a stable support, unlike people using NW poles during the march, because the construction of the poles slightly changes the body position while walking, compared to the classic NW. During the phase when the RSA pole is pressed, muscles perform additional work overcoming resistance of the elastic shock absorber in the pole. Pressing the shock absorber changes the length of the stick, which, when shortened to the maximum, reaches the length of a traditional NW stick. Releasing the pressure causes the stick to deform to its original length with the same force, which can give the impression of a change in the body balance. A detailedtechnical description of the Slimline 4 RSA poles that wereused for the intervention and the principles of the NW and RSA pole adjustment are described in Marciniak et al. [27].

### 2.5. Measurements

#### 2.5.1. Romberg Sharpened Test

To evaluate functional changes in body balance, the Tandem Romberg test (known as the Romberg sharpened test) was performed. This detects changes in body balance, depending on the level of proprioception in the participating subjects [2]. The interpretation of the test is analogous to that of the conventional Romberg test. According to Lee [33] and Herdman and Clendaniel [34], the test has 49–60% sensitivity, 95% specificity and a prediction of 82% [33].

The test subject should stand barefoot with arms crossed in front of the chest. This is to determine which lower limb is dominant, for example by giving the following command “please kick some imaginary object in front of you with your leg”. Then the subject stands with the foot of the dominant leg in the front and the other foot with the toes just behind the heel in one line.

The feet should be perfectly aligned so as not to form an angle (Figure 3).

Initially, the subject stands with the eyes open (EO) and vision set on a point 1 meter away, then remains in the same position with the eyes closed (EC).

During the test, people were belayed by the researcher, and to ensure maximum safety for the participants, gym mattresses were placed around the site.

The result obtained was the number of seconds during which the subject held the position without losing balance, allowing small oscillations [2]. The maximum measurement time was assumed to be 30 s.

All the subjects passed the EO test and achieved maximum results. In the further analysis, the results from EC test were analyzed, and the results of the EO test were not considered.

#### 2.5.2. Posturography Measurements

The research was conducted based on the measurement of the center of pressure (COP) displacement using the AccuGait ™ force platform (AMTI PJB-101, Watertown, MA) with the Balance Trainer software, which is often used in neurology and in the examination of the elderly [35,36].

The platform was equipped with strain gauges that facilitated monitoring of the changes in ground reaction forces. The device was placed on a hard and flat floor surface. Based on the data collected by the platform, it is possible to estimate the position and deflection of the foot pressure center. The evaluation was carried out twice, before and after the intervention.

Raw data were collected with frequency of 100 Hz and then low-pass filtering was done to remove noise from the obtained signal of COP displacements—the fourth-order Chebyshev II filter with cut-off frequency of 10 Hz was used [37].

During the examination, the subjects were asked to take an upright, vertical position, stand still with the upper limbs alongside the body, remaining silent—no conversation was allowed. The feet were placed in a position similar to their natural stance—heels in one line, feet positioned at an angle of about 30–40° to each other, with a 5 cm gap between the heels [31]. Before starting the study, an outline of each participant’s feet was made, which allowed for a similar positioning of the subject during subsequent trials.

In this position, three 30-sposturographic analyses were consecutively performed. They were separated by two 2-min breaks:-standing with the eyes open (EO),-standing with the eyes closed (EC),-testing body stability under dual task (DT) conditions.

DT—while standing, counting down every 3 numbers, starting from 200 [5].

The body balance assessment took up to 15 min including rest breaks.

During the test, belaying was applied and, if necessary, the subjects could rest in a sitting position.

To avoid the sequence being memorized, the trials were carried out at random. Each trial was run twice, and the results were averaged. In the previous work, it was found that averaging the results from two measurements allowed to obtain ICC (inter-class correlation) at a level above 0.9 in the case of the average displacement velocity COP [38].

### 2.6. Outcomes

The primary outcomes of the study were:(a)number of seconds during which the subject held the position without losing balance in Romberg test;(b)average velocity of COP displacements (and its components in anterior-posterior and medio-lateral directions), as well as ellipse area of 95% in posturographic measurements.

The study population was characterized by age, body weight and height as well as BMI (calculated as body weight/height^2^) as secondary outcomes.

### 2.7. Sample Size and Randomization

Taking into account one-factor effect of time in repeated measures (lack of data connected to effect size for two-factors study structure “group”x”time” in case of two different marching training), range of number of subjects is estimated from 13 to 22 with power statistics = 0.8 and alpha level = 0.05.

Women qualified for the study were randomly assigned to one of two groups with use of randomization generated using Excel software (first step—creation of column with randomized numbers in range 0–0.99 for each participant; second step—assignment to groups EG and CG in next column with function: if (value < 0.5;“EG”;“CG”)).

### 2.8. Statistical Analysis

The main calculations of dependent variables were based on two-way ANOVA (*F*-test) analysis methods. An analysis was made with repeated measurements before and after training (“time” factor with two levels—pre and post), and with two levels of an intergroup factor (“group”—EG and CG). For interaction effects (“time × group”) and main effects (“time” and “group”), the eta-squared effect size was calculated. The effect size indicates the percent of variance explained by particular effects of the dependent variable. To compare the average values of primary outcomes (both within and between groups) Bonferroni correction post hoc comparisons were conducted.

Statistical analyses were computed using Statistica v. 13.0 software (TIBCO Software Inc., Palo Alto, CA, USA). Statistical significance was defined as *p* < 0.05.

## 3. Results

Five participants did not complete the eight-week intervention period, including one person who dropped out due to injury, two due to illness, and two for undisclosed personal reasons. The remaining 35 people completed the training sessions with the attendance maintained above the required 80%. Finally, the effect of the intervention was assessed in 34 women (EG = 17, CG = 17)—one person did not take part in tests after intervention program without giving any reasons (Figure 4).

Basic characteristics of both groups are presented in Table 1.

Significant between-groups differences in body weight wereobserved. Average values of age, body height and BMI were similar in both groups.

In the case of the Sharpened Romberg test, a statistically significant “time” effect was found (Figure 5). Compared to the baseline, participants improved their scores after the intervention (F = 18.62, *p* < 0.001, ɳ^2^ = 0.37, power analysis = 0.98). Post hoc analysis revealed statistically significant pre-post differences for EG and CG (*p* = 0.034 and *p* = 0.022, respectively).

In the case of the posturography test (Table 2) none of the statistically significant “time × group” effects was found for any of the parameters tested (*p* > 0.05) under all study conditions (eyes open, eyes closed, and dual-task).

There were no statistically significant main effects for the “group” factor (*p* > 0.05) or for detailed between-groups comparisons, either.

In the case of the test with the eyes closed for the area of 95% of COP displacements, a significant value of the repeated measurement factor “time” was recorded (F = 4.20, *p* < 0.05, ɳ^2^ = 0.12). This result was related to slightly higher results in the CG group after training (within-group difference was not significant in post hoc analysis).

## 4. Discussion

The goal of this study was to determine the impact of marching with poles with an integrated resistance shock absorber and classic poles on improving balance in women. To the best of our knowledge, this study is the first to compare the effects of RSA and NW on body balance in healthy women.

In previous studies, the effect of NW on body balance compared to resistance exercises was evaluated [18,39,40]. However, no studies have been undertaken so far to assess the effects of walking with additional resistance from the RSA cushioning system on improving the static balance in senior females. It was assumed that workout consisting of walking with the use of poles has the value of a motor-cognitive intervention and provides participants with balance training in the conditions of performing a double task, as the participants walk while talking with each other, which requires information to be processed along with maintaining the balance and without changing the rhythm and walking speed. Another assumption made was that walking with RSA poles causes the body to work in unstable support conditions with additional resistance, compared to the work performed by the body during the classic NW. Both of these factors provided an additional stimulus for the marching women.

Research on the influence of NW training on the static balance in seniors gives results that are varied and difficult to interpret. In the Lajoie [41] study, after the same period of time (8 weeks, 16 sessions) there was no difference in postural sway observed, either. Moreover, the results showed no significant change between the groups. On the other hand, a statistically significant reduction in reaction time was observed in the elderly while standing, compared to the control group, and the difference remained significant even after a two-week retention period. Lee and Park [23] noticed a significant improvement in one-leg stance after 12 weeks of NW training. Kocur et al. [25] after 12 weeks of the NW training showed a significant improvement in static balance in the group ofwomen, which was measured by the Forward Reach Test (FRT) and Upward Reach Test (URT) by standing on a balance training platform with a balancing measurement function. However, Gobbo et al. [42], after analyzing eight experiments that met the assumed eligibility criteria, did not indicate any potential relationship between exercise and the improvement of static and dynamic balance during a dual task performed by healthy seniors.

Maranhão-Filho et al. [2] demonstrated that healthy elderly people are able to stand with EC without falling for at least 30 s. In the case of healthy, physically fit females, the standard posturographic tests used in this study proved too easy to demonstrate changes that probably occurred in the system of maintaining postural stability. Moreover, it is noteworthy that the examined women were younger than in other studies in which the influence of NW on balance was examined and where significant differences were obtained [20,23].

In the case of posturographic research, we expected to see differences in effects between groups, but there was no statistically significant interaction effect or the main “time” effect in most variables. Additionally, there were no other significant tendencies in the intergroup differentiation.

Only a significant value of the factor of the repeating “time” measurement was recorded, in the case of the EC test for the Area of 95% of COP displacements. It was connected to higher results in CG after training. It is possible that the result obtained is just an ‘artefact’ related to the relatively lower reliability of these parameters in comparison to the average velocity of COP displacement [43].

Despite the lack of intergroup differences in BMI, women in the EG group were characterized by higher weight, which could have influenced the results. In studies on the influence of obesity on body postural stability [44,45], higher oscillation values were observed in people with higher body weight; therefore, higher body weight values in women with EG could have affected the overall result of the study. On the other hand, in the studies of Carral et al. [46], it was found that postural control in obese elderly people may depend on the amount of PA performed.

The results of the Romberg sharpened test showed the statistically significant main effect, which is “time”, for both groups. The tandem stance of the feet in this test makes it difficult to maintain a stable posture, as opposed to standing in a relaxed position. The reduction in the field of support may activate the hip joint strategy, which is one of the three major stability maintenance and recovery reactions. According to Horak [47], the use of this strategy may also be associated with a reduced or unstable support plane. In a similar test, Marciniak et al. [27] (after 8 weeks, 16 sessions) observed enhanced agility, endurance and muscle strength. Therefore, improving participants’ overall performance as a result of the experiment may explain the results obtained in the functional test. According to Maranhão-Filho et al. [2], Romberg’s test illustrates the level of proprioception, which may indicate a positive impact of the experiment on the proprioceptive balance control in the elderly.

The results of the Romberg test could be related to the potential improvement in strength and endurance, which were not significant in the posturographic examination. They could stem from the specificity of the examination itself: a reduced support plane while standing and a strategy of maintaining balance other than in the posturographic examination.

The final result of the study could have been influenced by relatively short time during which the intervention took place (8 weeks and 16 training sessions). The time in which the study was conducted marked the period between the holidays. Further attempts to continue could cause a change in the participants’ behavior resulting from the traditional involvement of women in holiday preparations, and a change in their eating habits.

## 5. Limitations

Finally, it should be mentioned that the study has some limitations. First, it only includes a group of women. Studies involving both men and women would provide stronger general evidence. Secondly, the training was relatively short. Perhaps continuing training would also result in significant differences in all tests used. On the other hand, in other studies, eight-week periods of training made it possible to observe significant changes in physical fitness among the elderly. Next, during each session, we focused on the technical details of the march with poles, because proper execution of exercises and walking technique were very important in our experiment. However, a large number of people participated in the training at the same time, and the marching technique may not always have been correct. Perhaps more personalized training should be used to ensure better control of exercise execution in the next study. The very low effect of applied exercise on results obtained in posturography tests is also interesting. As stated in the discussion, perhaps the tests that were used were too simple, and the ceiling effect was met.

## 6. Conclusions

The training sessions and body balance tests did not allow any significant impact of the type of poles on the change in the body balance using posturographic tests to be demonstrated; and they seemed to have a similar effect on the results obtained in functional tests.

Taking into account the limitations of the study, further research is necessary with extended intervention times and more difficult tasks in posturographic examination.

On the other hand, when the base of support limitation was applied in the Romberg functional test (Romberg sharpened), the statistically significant main effect of “time” was found in both groups. Considering that the results of this test are linked to the possibility of detecting deficiencies in proprioception, they may indicate the beneficial effect of walking on the static balance inwomen. Compared to the baseline, both groups improved their results, which may indicate an improvement in perception and processing of sensory information. This may be the result of the intervention—regardless of the type of training applied.

## Figures and Tables

**Figure 1 healthcare-09-00267-f001:**
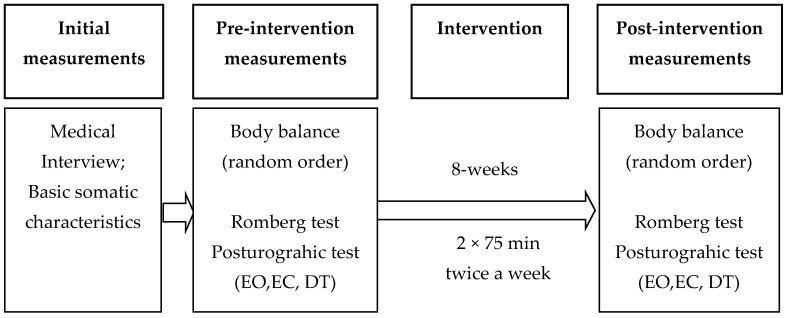
General overview of the experiment. Note: EO—Eyes Open, EC—Eyes Open, DT –Dual Task.

**Figure 2 healthcare-09-00267-f002:**
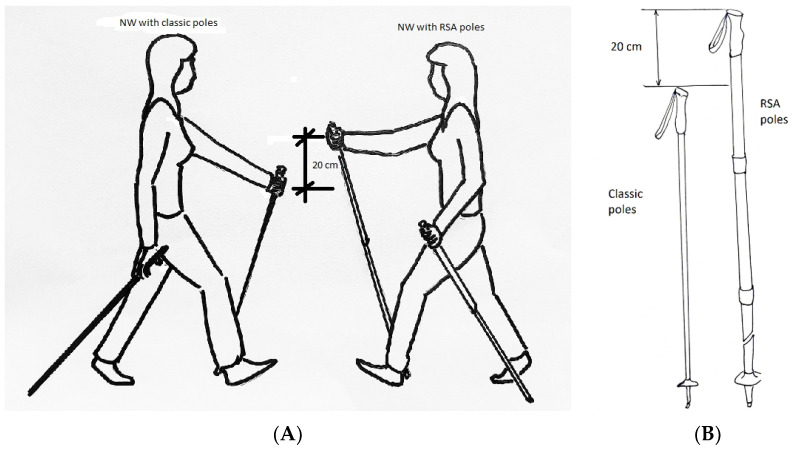
Correct position of upper limbs during Nordic Walking with classic and resistant shock absorber (RSA) poles (**A**), and differences between poles (**B**).

**Figure 3 healthcare-09-00267-f003:**
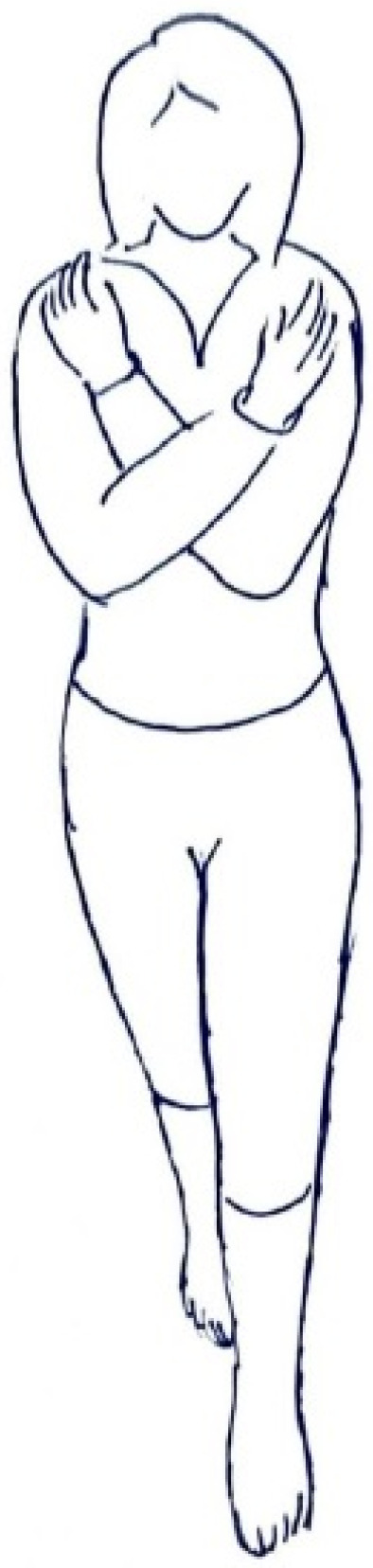
Correct positioning of the feet “heel to toe” in the Romberg sharpened test—tandem stance (the dominating lower limb at the front position).

**Figure 4 healthcare-09-00267-f004:**
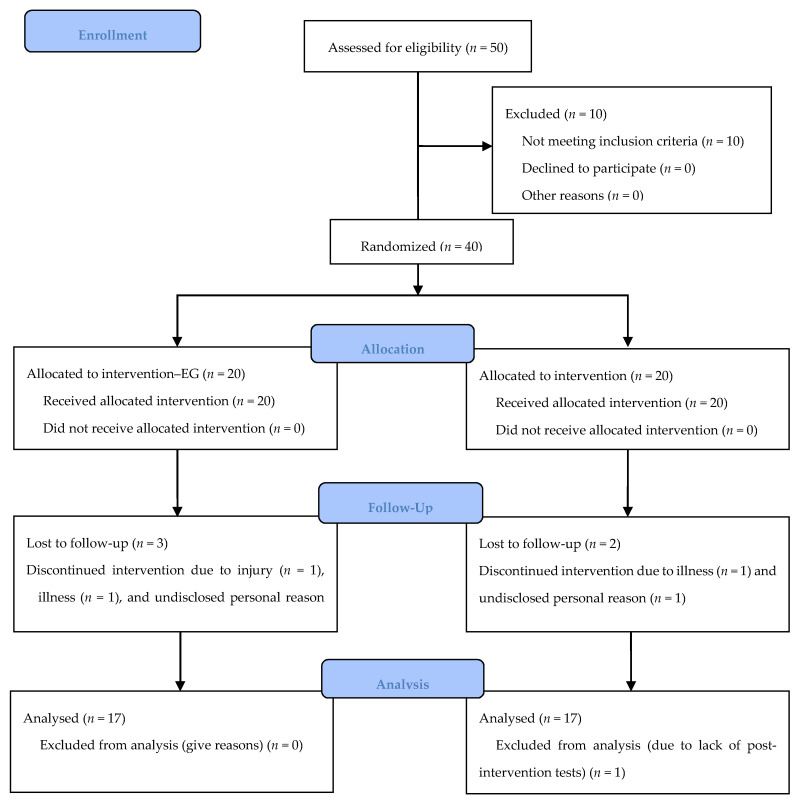
Flowchart of the experiment participants.

**Figure 5 healthcare-09-00267-f005:**
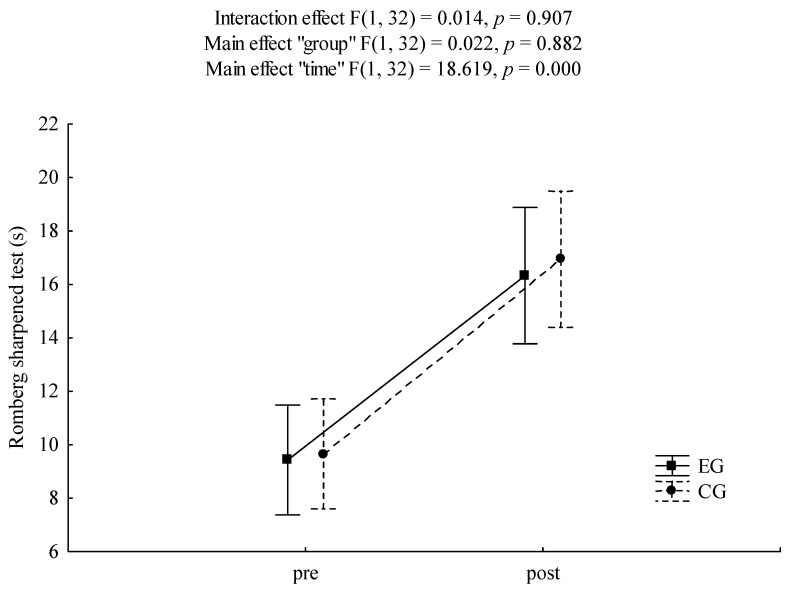
Mean values, standard error of measurements andanalysis of variance ANOVA for results of Romberg sharpened test. EG—experimental group, CG—control group.

**Table 1 healthcare-09-00267-t001:** Mean values, standard deviations and results of between-group comparison (*t*-test for independent samples) for basic characteristics of participants prior to the start of the experiment.

	EG (*n* = 20)		CG (*n* = 20)		t (*p*)
Measure	M	SD	M	SD	
Age (years)	67.71	4.16	65.88	4.09	1.29(0.207)
Body height (cm)	161.94	4.04	159.25	5.45	1.64(0.112)
Body weight (kg)	74.31	11.17	65.63	11.19	2.26(0.030)
BMI (kg/m^2^)	28.31	3.90	25.90	4.53	1.66(0.106)

BMI—body mass index; EG—experimental group, CG—control group.

**Table 2 healthcare-09-00267-t002:** Mean and standard deviation values for body balance for Eyes Open, Eyes Closed and Dual Task measures results of analysis of variance.

	Pre		Post							
Variable	M (sd) EG	M (sd) CG	M (sd) EG	M (sd) CG	Interaction *F*(*p*)	ɳ^2^	Group *F*(*p*)	ɳ ^2^	Time *F*(*p*)	ɳ ^2^
EO										
Vcop [cm/s]	1.14(0.46)	0.99(0.31)	1.17(0.43)	1.04(0.26)	0.03(>0.05)	0.00	1.41(>0.05)	0.04	0.62(>0.05)	0.02
VcopY [cm/s]	0.95(0.46)	0.83(0.29)	0.96(0.41)	0.81(0.20)	0.23(>0.05)	0.01	1.33(>0.05)	0.04	0.01(>0.05)	0.00
VcopX [cm/s]	0.45(0.20)	0.39(0.12)	0.48(0.20)	0.48(0.20)	0.87(>0.05)	0.03	0.26(>0.05)	0.01	3.53(>0.05)	0.10
A95 [cm^2^]	1.71(0.68)	1.98(1.11)	2.10(1.44)	2.22(1.44)	0.12(>0.05)	0.00	0.32(>0.05)	0.01	2.16(>0.05)	0.06
EC										
Vcop [cm/s]	1.55(0.69)	1.36(0.45)	1.59(0.64)	1.46(0.69)	0.12(>0.05)	0.00	0.63(>0.05)	0.02	0.86(>0.05)	0.03
VcopY [cm/s]	1.34(0.67)	1.19(0.41)	1.35(0.62)	1.21(0.60)	0.00 (>0.05)	0.00	0.62(>0.05)	0.02	0.07(>0.05)	0.00
VcopX [cm/s]	0.53(0.27)	0.46(0.18)	0.58(0.27)	0.58(0.34)	0.56(>0.05)	0.02	0.13(>0.05)	0.00	3.42(>0.05)	0.10
A95 [cm^2^]	1.91(1.10)	2.08(1.11)	2.33(1.15)	2.64(1.75)	0.09(>0.05)	0.00	0.41(>0.05)	0.01	4.20(<0.05)	0.12
DT										
Vcop [cm/s]	1.63(0.91)	1.50(0.81)	1.54(0.57)	1.64(0.83)	1.23(>0.05)	0.04	0.00(>0.05)	0.00	0.06(>0.05)	0.00
VcopY [cm/s]	1.29(0.60)	1.21(0.64)	1.26(0.48)	1.30(0.69)	0.66(>0.05)	0.02	0.01(>0.05)	0.00	0.16(>0.05)	0.00
VcopX [cm/s]	0.73(0.60)	0.65(0.41)	0.65(0.26)	0.74(0.42)	1.55(>0.05)	0.05	0.00(>0.05)	0.00	0.00(>0.05)	0.00
A95 [cm^2^]	3.32(3.79)	3.40(4.57)	2.84(1.96)	5.12(6.41)	3.52(>0.05)	0.10	0.69(>0.05)	0.02	1.13(>0.05)	0.03

EG—experimental group; CG—control group; Vcop—velocity; VcopY—anterior-posterior; VcopX—medio-lateral; A95—area 95 percentile.

## Data Availability

The data supporting reported results are available in the corresponding author.

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
