# Peer review of "The Effect of Nordic Walking Training with Poles with an Integrated Resistance Shock Absorber on the Body Balance of Women over the Age of 60"

_healthcare, 2021, doi:10.3390/healthcare9030267_

Round 1
Reviewer 1 Report
I would like to thank the authors for a very interesting article. Although the following changes are advised:
Introduction
Lines 32-33: This phrase is not supported by any reference. Please include.
Lines 49-50: In this sentence you talk about previous studies that have shown the effects of physical training on gait and body balance, but you only cites one study. Please add more works that have proven it.
Lines 60-67: This paragraph or sentences have not been referenced.
Methods
They must specify the type of study they have carried out.
Was the clinical trial registered? All clinical trials must be registered.
Add a sample size calculation, with reference to the literature and with enough detail to allow replication.
Add a section that explains the distribution of the sample in the two groups, how it was done, if it was random ...
Results and Discussion
These sections are carried out correctly, even if the hypothesis raised has not been achieved.
References
This section must be reviewed in its entirety. There must be a homogeneity.
Reviewer 2 Report
Thank you for the opportunity to review the paper. I understand and agree with the scientific basis of the study, and believe the results fill gaps in the literature. I have a few recommendations for the authors to address before publication. Among those, the most important is for them to address the issue of English writing. There is awkward phrasing throughout the document. I recommend that the authors either overhaul their writing approach or engage the services of a professional editor.
The following points are all minor comments:
- In line 19-20, it is not immediately clear what a significant effect of "time" is. I understand that it's the main effect observed in the ANOVA, but since the abstract is the first thing that most readers read, please rephrase to make it clearer.
- There needs to be a clearer explanation of what exactly the inclusion criteria is. What does "free from diseases" mean, exactly? What level of provider conducted the medical examination for study inclusion?
- Line 110: Where is the "80%" referenced from?
- Figures 1 and 2 need to be reformatted to be neatly presented
- Table 2 needs to be reformatted. It was really hard to read.
- Line 134 - cite endomondo.
- Lines 139 to 150 - cite the basis of these exercises. Were they previously used in other studies by other authors?
- Line 257 - by convention, if p = .05, then it is not significant. shouldn't be p < 0.05 in the test?
Reviewer 3 Report
This is a very interesting study with an important impact on the rehabilitation area.
My comments are listed below:
The rationale for using the pole absorb system for improving the balance score is not adequately presented.
The study lucks power analysis. Authors need to run a posthoc power analysis in order to assess the power of their analysis.
The authors need to include a strength and weakness section for presenting the originality of the study as well as the pitfalls or weaknesses detected by the authors.
Reviewer 4 Report
This study is quite interesting and I would like to suggest some modifications:
first, modify the first part of methods, and not not refer this manuscript as a part of the prior one;
add a sample size test power;
I would like to ask if the authors could iprove their experimental figure; There are some manuscripts with a better fig. design;
At line 139, please describe the exact used exercises (and not examples) to make the session more concisely;
Maybe in fig 3, you can provide a fig more detailed, regarding the differences between the two RSAs
Fig 4, add a draw instead fo a picture. Please, ensure that the fig is self-explained.
Change statistical methods to statistical analysis;
check the format of table 2;
Reviewer 5 Report
Authors present a randomized controlled trial aiming to identify if NW empower by RSA poles may improved balance compared to classic NW. The trial is well-designed and interesting, and authors need to be congratulated for their efforts. Otherwise, I believe that several changes should be made before publication, mainly concerning statistical analysis.
At first, I believe that authors need to uniform their paper according to the CONSORT guidelines for randomized trials. Once performed, this may really help the reader.
Abstract: please be consistent in the use of “.” or “,” for decimals number here and throughout the manuscript.
Please also avoid decimals when they cannot be measured (e.g. age, or height, etc.).
Methods section may contain only methods, whereas all the presentation of the population, the drop outs and their reasons, really need to be moved to the results section.
Please explain (line 112) the sentence.
Please give clear inclusion and exclusion criteria. Please specify the randomization methods, the possible blocking sequence.
Please also state the primary outcome, the secondary outcomes, and if any blinding was applied.
Please uniform the Flow diagram to the Consort statement, and give reason for drop out subjects.
Statistical methods need to be improved. At first, authors need to verify the data distribution. Secondly, a baseline comparison of the two groups for anthropometric and outcomes needs to be performed.
Along with the two-way ANOVA (that it is not correct in the data are nonnormally distributed), a within-group and between-group classic comparison (e.g. paired-samples t-test or independent samples t-test, respectively) should be useful.
Please re-format the tables that are absolutely a mess. They are not readable.
I suggest to state the study limitations before the conclusion, and not vice versa.
Please uniform the references according to the journal format. They are not consistent.
Round 2
Reviewer 1 Report
The authors have answered correctly to all the questions raised, giving a higher quality to the study.
Reviewer 3 Report
The authors have responded adequately to the majority of my comments. No further comments. Accept in the current form.
Reviewer 5 Report
Authors have addressed all my comments